# Bone supression in planar X-ray images with Stable Diffusion

**Daniel Sanderson**[1,2] (iD)                                   DSANDERS@ING.UC3M.ES

**Mariana Elizalde**[1]                                          MELIZALD@PA.UC3M.ES

**Paula Ochotorena**[1]                                          POCHOTOR@PA.UC3M.ES

**Manuel Desco**[1,2,3,4]                                        MANUEL.DESCO@UC3M.ES

**Monica Abella**[1,2,3]                                         MABELLA@ING.UC3M.ES

[1] *Departamento de Bioingeniería, Universidad Carlos III de Madrid. Madrid, Spain*

[2] *Instituto de Investigación Sanitaria Gregorio Marañón. Madrid, Spain*

[3] *Centro de investigación en red en salud mental (CIBERSAM), Madrid, Spain*

[4] *Centro Nacional de Investigaciones Cardiovasculares Carlos III (CNIC), Madrid, Spain.*

**Editors:** Accepted for publication at MIDL 2025

## Abstract

Bone suppression is a processing technique that aims to enhance the visualization of chest radiographic images by attenuating bones while preserving soft tissue details. This has been achieved with deep learning methods but they either introduce blurring or do not fully remove bones. In this work, we propose a bone removal method for radiography based on SD. To address the lack of publicly available bone suppression datasets, the model is pretrained using a synthetic dataset simulated from computed tomography scans. Preliminary evaluation demonstrates the ability of the proposed model to fully remove bones while preserving spatial resolution.

**Keywords:** X-ray, radiography, bone supression, diffusion models, SD

## 1. Introduction

X-ray imaging is the prefered modality for lung diagnosis either with chest radiography CXR or computed tomography (CT) scans. Although CXR requires lower radiation doses and shorter acquisition times and is more cost-effective, it shows a limited diagnostic value due to anatomical superposition, especially bone tissue, making it difficult to detect lesions. This can be alleviated by bone-suppression techniques, which can be divided in two groups. The first is Dual Energy (Zarshenas et al., 2019; Yang et al., 2017; Blau and Michaeli, 2017), which involves acquiring two X-ray images at different voltage levels (Johnson, 2012), thus doubling the radiation dose delivered to the patient (Rajaraman et al., 2022). The second is based on image processing techniques, particularly deep learning-based algorithms, which use a single image facilitating their clinical application while reducing dose (Sirazitdinov et al., 2020). Several approaches to bone suppression have been explored in the literature (Zhou et al., 2020; Rajaraman et al., 2021; Cho et al., 2022). However, in general they do not fully remove bones and lead to blurry results. This could be solved by using diffusion models, which have shown stunning results in the generation of high-resolution images in

the medical field. To avoid having to train the diffusion model from scratch, pre-trained foundational models such as Stable Diffusion (SD) can be fine-tuned for specific applications.

In this work, we explore the utility of SD to achieve high-resolution bone suppression images. Given the scarcity of Dual Energy public databases, we propose a novel strategy for generating bone supressed images based on the simulation of 2D projections from 3D volumes.

## 2. Materials and methods

### 2.1. Dataset

A simulated database was generated using 48 chest CT volumes in DICOM format obtained from the Radiology Department of the Hospital General Universitario Gregorio Marañón, 32 from the MIDRC portal (NIBIB), and 157 from the CT-RATE dataset (Hamamci et al., 2024). Two projections were obtained for each CT volume: the standard CXR and the bone-suppressed one. To transform the lying position of the patient in the CT to the standup position in the CXR, the patient bed was removed from the CT by segmenting the patient slicewise with a 2D UNet. A bone mask was obtained by thresholding, and bone supressed CTs were obtained by replaicng the voxels in the bone mask with a value of 1.12 $g/cm^3$ to emulate soft tissue. CXRs were simulated using FUX-Sim (Abella et al., 2017), with the standard geometry for chest radiography and a source spectrum of 110 KV generated with Spektr (Siewerdsen et al., 2004). Projection images were downsampled to a matrix size of 512x512 (pixel size of 0.8x0.8 mm) which is the size expected by the SD 2.0 base model. CT scans were rotated within a range of $\pm20$ degrees to simulate variations in the position of the patient during acquisition, resulting in 9,515 images for training and 797 for validation. Evaluation was performed with a test set that consisted of 20 radiographs from 20 patients acquired at 0 degrees.

### 2.2. Proposed method

The proposed method is built in two stages. In the first stage, SD was trained to generate bone-suppressed CXR images from a text prompt. Rather than fine-tuning the entire network, we used Textual Inversion as in (de Wilde et al., 2023) to avoid catastrophic forgetting and reduce computational cost (Gal et al., 2022). The entire model was frozen except for a text embedding matrix, which was trained to accurately depict the concept corresponding to a token representing bone-suppressed images. In the second stage, for SD to learn to generate bone supressed images from CXR, we combined it with a ControlNet (Zhang et al., 2023). The ControlNet was built using the same architecture as SD. The feature maps were added at each layer of the combined model. The weighting of the addition was controlled by zero convolutions, which prevent the ControlNet from degrading the results of SD during the initial stages of training, gradually increasing its contribution as training progresses. Zero convolutions were applied to all of the encoder and decoder blocks of the network to increase the expresivity of the combined model. Although generally different prompts are used to train the ControlNet, we only used the token learnt in the first stage given that it is the only one representing the concept of bone-supressed CXRs.

Training was performed on a single GeForce RTX 3090 GPU of 24 GB for 14 hours. For Textual inversion, only 100 bone-supressed images were used, as in (de Wilde et al., 2023). Inference was done for 100 sampling steps, with a guidance value of 2 and a ControlNet conditioning weight of 1. To increase inference speed, the UniPC scheduler was used (Zhang et al., 2023). Evaluation was done after enhancing the images with CLAHE, using a size tile of 70 píxels and a clip limit of 0.0001, and with a Laplacian Pyramid of 3 levels. The proposed method was compared with a supervised UNet, which consisted of a ResNet34 encoder and was trained with a MSSIM loss.

## 3. Results and Discussion

Figure 1 shows that the vanilla UNet was unable to completely remove the bones and blurred the trachea and some of the bronchi. In contrast, SD removed the bones entirely while preserving spatial resolution, resulting in a better perceptual quality reflected by the LPIPS and CPBD metrics (LPIPS is 0.11 and 0.08, and CPBD is 0.37 and 0.48, for the UNet and SD respectively). However, in some regions of the lungs, hallucinations of small high-density structures were introduced that could potentially limit the diagnostic value of the image. This is reflected by lower values of MSSSIM and PSNR compared to the UNet (MSSIM is 94.32 and 91.96, and PSNR is 31.341 and 29.53, for the UNet and SD respectively) This effect could be mitigated by training with a larger dataset. Inference took less than 15 seconds, allowing its real-time application. For this method to be applicable to real radiographies, in the future we will simulate radiographies of larger sizes and train SD on patches of fixed size.

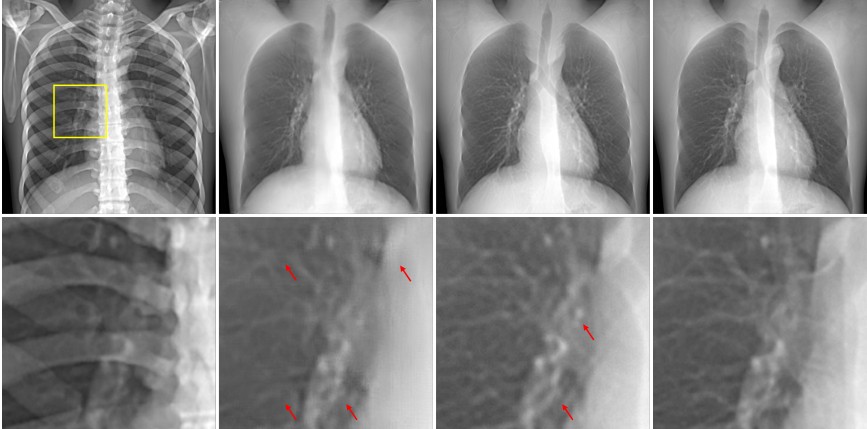

Figure 1: From left to right: CXR, UNet, proposed method, bone-supressed CXR. The second row corresponds to a zoom of the region indicated by the yellow rectangle. Red arrows indicate hallucinations or loss of anatomical structures.

To conclude, results showed the potential of the proposed method for bone supression without the loss of spatial resolution or uncomplete bone removal of previous methods.

## Acknowledgments

This work was funded by Instituto de Salud Carlos III (ISCIII) through the projects DTS22/00030 and PI23/01181, co-funded by the European Union, and through projects PMPTA22/00121 and PMPTA22/00118, co-funded by the European Union NextGenerationEU, Mecanismo de Recuperación y Resiliencia (MRR). The CNIC is supported by Instituto de Salud Carlos III, Ministerio de Ciencia e Innovación, and the Pro CNIC Foundation.

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
