# OpenReview forum: "Bone supression in planar X-ray images with Stable Diffusion"
_MIDL.io/2025/Short_Papers — MIDL 2025 - Short Papers_

### Official Review · Reviewer_PAPs · 2025-04-20

**Rating:** 4
**Confidence:** 4

**Summary:**

This paper uses Stable Diffusion for bone suppression in synthetic chest radiography. Compared to results generated by U-Net, the authors find that Stable Diffusion produces better perceptual quality but lower PSNR.

**Strengths:**

1. The problem is well-defined.
2. The use of Stable Diffusion is reasonable.

**Weaknesses:**

1. It is not entirely clear why the authors chose Textual Inversion and ControlNet in their framework. The motivation is not well-explained.
2. I foresee that the trained model may struggle when applied to real chest X-rays.

---

### Decision · Program_Chairs · 2025-05-01

Accept